DISCOVERY REPORT

# Genomic surveillance uncovers a pandemic clonal lineage of the wheat blast fungus

Sergio M. Latorre[1], Vincent M. Were[2], Andrew J. Foster[2], Thorsten Langner[2], Angus Malmgren[2], Adeline Harant[2], Soichiro Asuke[3], Sarai Reyes-Avila[2], Dipali Rani Gupta[4], Cassandra Jensen[5], Weibin Ma[2], Nur Uddin Mahmud[4], Md. Shabab Mehebub[4], Rabson M. Mulenga[6], Abu Naim Md. Muzahid[4], Sanjoy Kumar Paul[4], S. M. Fajle Rabby[4], Abdullah Al Mahbub Rahat[4], Lauren Ryder[2], Ram-Krishna Shrestha[2], Suwilanji Sichilima[6], Darren M. Soanes[7], Pawan Kumar Singh[8], Alison R. Bentley[8], Diane G. O. Saunders[5], Yukio Tosa[3], Daniel Croll[9], Kurt H. Lamour[10], Tofazzal Islam[4], Batiseba Tembo[6], Joe Win[2]*, Nicholas J. Talbot[2]*, Hernán A. Burbano[1]*, Sophien Kamoun[2]*

1 Centre for Life's Origins and Evolution, Department of Genetics, Evolution and Environment, University College London, London, United Kingdom, 2 The Sainsbury Laboratory, University of East Anglia, Norwich Research Park, Norwich, United Kingdom, 3 Graduate School of Agricultural Science, Kobe University, Kobe, Japan, 4 Institute of Biotechnology and Genetic Engineering, Bangabandhu Sheikh Mujibur Rahman Agricultural University, Gazipur, Bangladesh, 5 John Innes Centre, Norwich Research Park, Norwich, United Kingdom, 6 Zambia Agricultural Research Institute, Mt. Makulu Central Research Station, Lusaka, Zambia, 7 Department of Biosciences, University of Exeter, Exeter, United Kingdom, 8 International Maize and Wheat Improvement Center, (CIMMYT), Texcoco, Mexico, 9 Laboratory of Evolutionary Genetics, Institute of Biology, University of Neuchâtel, Neuchâtel, Switzerland, 10 Department of Entomology and Plant Pathology, University of Tennessee, Knoxville, Tennessee, United States of America

☙ These authors contributed equally to this work.
* Joe.Win@tsl.ac.uk (JW); Nick.Talbot@tsl.ac.uk (NJT); h.burbano@ucl.ac.uk (HAB); Sophien.Kamoun@tsl.ac.uk (SK)

The Editors encourage authors to publish research updates to this article type. Please follow the link in the citation below to view any related articles.

## Abstract

Wheat, one of the most important food crops, is threatened by a blast disease pandemic. Here, we show that a clonal lineage of the wheat blast fungus recently spread to Asia and Africa following two independent introductions from South America. Through a combination of genome analyses and laboratory experiments, we show that the decade-old blast pandemic lineage can be controlled by the Rmg8 disease resistance gene and is sensitive to strobilurin fungicides. However, we also highlight the potential of the pandemic clone to evolve fungicide-insensitive variants and sexually recombine with African lineages. This underscores the urgent need for genomic surveillance to track and mitigate the spread of wheat blast outside of South America and to guide preemptive wheat breeding for blast resistance.

## Introduction

Plant disease outbreaks threaten the world's food security at alarming levels [1,2]. Understanding pathogen evolution during epidemics is essential for developing a knowledge-based disease management response. Genomic surveillance allows for rapid and precise pathogen identification, tracing of outbreak origin and can guide preventive measures. It therefore adds a unique

**Data Availability Statement:** Genotyping and whole-genome sequencing data included in this article were released without restrictions as soon as they were produced through the OpenWheatBlast Community (https://zenodo.org/communities/openwheatblast). OpenWheatBlast collects research output datasets on wheat blast and encourages scientists to analyze and share them before formal publication. We list below the preprints that were shared through the OpenWheatBlast community and whose data were analyzed in this publication: - J. P. Ascari, et al. 2021. Multiplex amplicon sequencing dataset for genotyping the wheat blast fungus from the Minas Gerais state of Brazil. https://doi.org/10.5281/zenodo.4737375 - D. Croll. 2021. Whole-genome analyses of 286 Magnaporthe oryzae genomes suggest that an independent introduction of a global pandemic lineage is at the origin of the Zambia wheat blast outbreak. https://doi.org/10.5281/zenodo.4655959 - C. Jensen, et al. 2019. Rmg8 confers resistance to the Bangladeshi lineage of the wheat blast fungus. https://doi.org/10.5281/zenodo.2574196 - S. M. Latorre, et al. 2022. A curated set of mating-type assignment for the blast fungus (Magnaporthales). https://doi.org/10.5281/zenodo.6369833 - S. M. Latorre & H. A. Burbano. 2021. The emergence of wheat blast in Zambia and Bangladesh was caused by the same genetic lineage of Magnaporthe oryzae. https://doi.org/10.5281/zenodo.4619405 - B. Tembo, et al. 2021. Whole genome shotgun sequences of Magnaporthe oryzae wheat blast isolates from Zambia. https://doi.org/10.5281/zenodo.4637175 - B. Tembo, et al. 2021. Multiplex amplicon sequencing dataset for genotyping pandemic populations of the wheat blast fungus. https://doi.org/10.5281/zenodo.4605959 - V. Were, et al. 2021. Genome sequences of sixty Magnaporthe oryzae isolates from multiple host plant species. https://doi.org/10.5281/zenodo.4627043 - J. Win, et al. 2021. A pandemic clonal lineage of the wheat blast fungus. https://doi.org/10.5281/zenodo.4618522 The datasets and scripts generated during and/or analyzed during the current study are available in the Github repository: https://github.com/Burbano-Lab/wheat-clonal-linage under the DOI: https://doi.org/10.5281/zenodo.7590238.

**Funding:** This study was supported by the UK Biological Sciences Research Council (BBSRC) grants BBS/E/J/000PR9795 to SK and NJT, BBS/E/J/000PR9796 to NJT, BBS/E/J/000PR9798 to SK and NJT, BB/P023339/1 to NJT, BB/W008157/1 to SK, BB/W008300/1 to HAB, BB/R01356X/1 equipment grant to University College London, the Gatsby Charitable Foundation (https://www.gatsby.org.uk/) to SK and NJT, The Krishi Gobeshona

dimension to coordinated responses to infectious disease outbreaks and is central to the Global Surveillance System (GSS) recently proposed to increase global preparedness to plant health emergencies [3].

In wheat, yield losses caused by pests and diseases average over 20% [4]. Wheat is currently threatened by the expanding blast pandemic caused by the ascomycete fungus *Magnaporthe oryzae* (Syn. *Pyricularia oryzae*), a formidable and persistent menace to major grain cereals that could contribute to total crop failure [5]. The disease first appeared in 1985 in Brazil but has been reported in Bangladesh and Zambia over the last years, causing, for example, an average yield loss of 51% in the Bangladesh outbreak in 2016 [6]. The occurrence of wheat blast on three continents with climatic conditions highly conducive to its spread, is unprecedented and represents a very significant threat to global food security which is exacerbated by the unprecedented twin challenge of climate change [7] and armed conflicts [8] in major agricultural regions.

Wheat blast emerged in Brazil in 1985 following the wide-spread deployment of wheat genotypes carrying the *RWT4* resistance gene but lacking *RWT3* (*RWT4+/RWT3-*). These two resistance genes recognize the blast effectors PWT3 and PWT4, respectively. Thus, the deployment of RWT4+/RWT3- varieties facilitated host jumps of *M. oryzae* isolates carrying PWT3, but not PWT4 effectors from ryegrass (*Lolium* spp.) to wheat, which was followed by loss of function mutations in the PWT3 effector and subsequent spread to common wheat varieties [9]. An aggressive lineage of the wheat blast fungus that first emerged in Bangladesh in 2016 was previously traced to the genetically diverse South American population [10]. However, the genetic identity and origin of the causal agent of an African outbreak, first detected in Zambia in 2018, remains unknown [11].

Here, we show that the recent emergence of wheat blast in Asia and Africa was caused by a single clonal lineage of the wheat blast fungus closely related to South American isolates and that the outbreaks in Zambia and Bangladesh originated by independent introductions. Through rapid genome analyses, we revealed that the disease resistance gene *Rmg8* as well as strobilurin fungicides are effective against isolates of the pandemic clonal lineage and confirmed our predictions in laboratory experiments. However, the pandemic lineage has the capacity to develop fungicide resistance and can mate with local finger millet (*Eleusine coracana*) blast fungus, highlighting the evolutionary potential of the African outbreak to cause further damage to wheat production across the continent. Our results demonstrate that genomics can rapidly identify emerging pathogen genotypes to guide disease management and counteract emerging pathogen lineages. These findings will inform management strategies for this devastating wheat disease and warrant further genomic surveillance to prevent and manage future outbreaks.

## Results and discussion

To determine the relationship between African wheat blast isolates from Zambia with populations from South America and Bangladesh, we selected 84 single nucleotide polymorphisms (SNPs) mined from the sequence data of Islam and colleagues [10] to discriminate between the Bangladesh lineage from other *M. oryzae* genotypes. We genotyped 537 *M. oryzae* samples from different geographical regions and hosts based on multiplex amplicon sequencing (MonsterPlex; see Material and methods) (*N* = 237) and publicly available genomes (*N* = 351) (Figs 1 and S1 and S1 Table). Using the set of isolates from which we genotyped the 84 SNPs and also sequence their whole genomes, we showed that the set of 84 Monsterplex SNPs accurately reflects the patterns of genome-wide diversity and host specificity of the blast fungus (S2 Fig). The Zambian isolates (*N* = 13, 2018 to 2020) are identical for the 84 SNPs to wheat blast

Foundation (KGF) of Bangladesh grants KGF TF50-C/17 and TF 92-FNS/21 to TI, The Leverhulme Trust (Philip Leverhulme Prize) to HAB, Royal Society RSWF\R1\191011 to HAB, and European Research Council BLASTOFF grant 743165 to SK. The funders had no role in study design, data collection and analysis, decision to publish, or preparation of the manuscript.

**Competing interests:** We have read the journal's policy and the authors of this manuscript have the following competing interests: KL is a founder of Floodlight Genomics, TI receives funding from Krishi Gobeshona Foundation of Bangladesh, and SK receives funding from industry and has filed patents on plant disease resistance.

**Abbreviations:** GSS, Global Surveillance System; LD, linkage disequilibrium; ML, maximum likelihood; PCA, principal component analysis; SNP, single nucleotide polymorphism; QD, Quality-by-Depth.

isolates from Bangladesh (*N* = 71, 2016 to 2020) and one genotype B71 from South America (Bolivia, *N* = 1, 2012). We conclude that this "B71 lineage," which emerged in Asia in 2016 and traces its origins to South America, is now established in Zambia.

We prioritized samples for whole-genome sequencing based on our genotyping analyses and combined the samples with existing datasets to generate a set of 71 whole-genome sequences of *M. oryzae* wheat isolates (*Triticum* lineage) from South America (*N* = 37), Asia (*N* = 21), and Africa (*N* = 13) (S2 Table). To gain insight into the phylogenetic relationship of the B71 lineage to other wheat isolates, we first performed unsupervised clustering of the 71 genomes using principal component analysis (PCA) based on pairwise Hamming distances (Fig 2A) and hierarchical clustering based on *f3*-outgroup statistics (S3 Fig). The B71 lineage shows reduced genetic diversity in comparison with South American isolates although incipient sub-structuring can be noted between Zambian and Bangladeshi clusters (Fig 2A, inset).

We decided to test the hypothesis that the B71 cluster is a clonal lineage and challenged it by measuring pairwise linkage disequilibrium (LD) (Figs 2B and S4). Unlike the South American population, the B71 cluster displays no patterns of LD decay, which is consistent with clonality [12,13] (Figs 2B and S4–S6).

We performed phylogenetic analyses to further define the genetic structure of the B71 clonal lineage. Owing to the much finer resolution obtained with genome-wide variation, we found that the Zambian and Bangladesh isolates clustered in separate well-supported clades with distinct phylogenetic affinities to South American isolates (Fig 2C). We conclude that the clonal lineage has spread to Asia and Africa through at least two independent introductions,

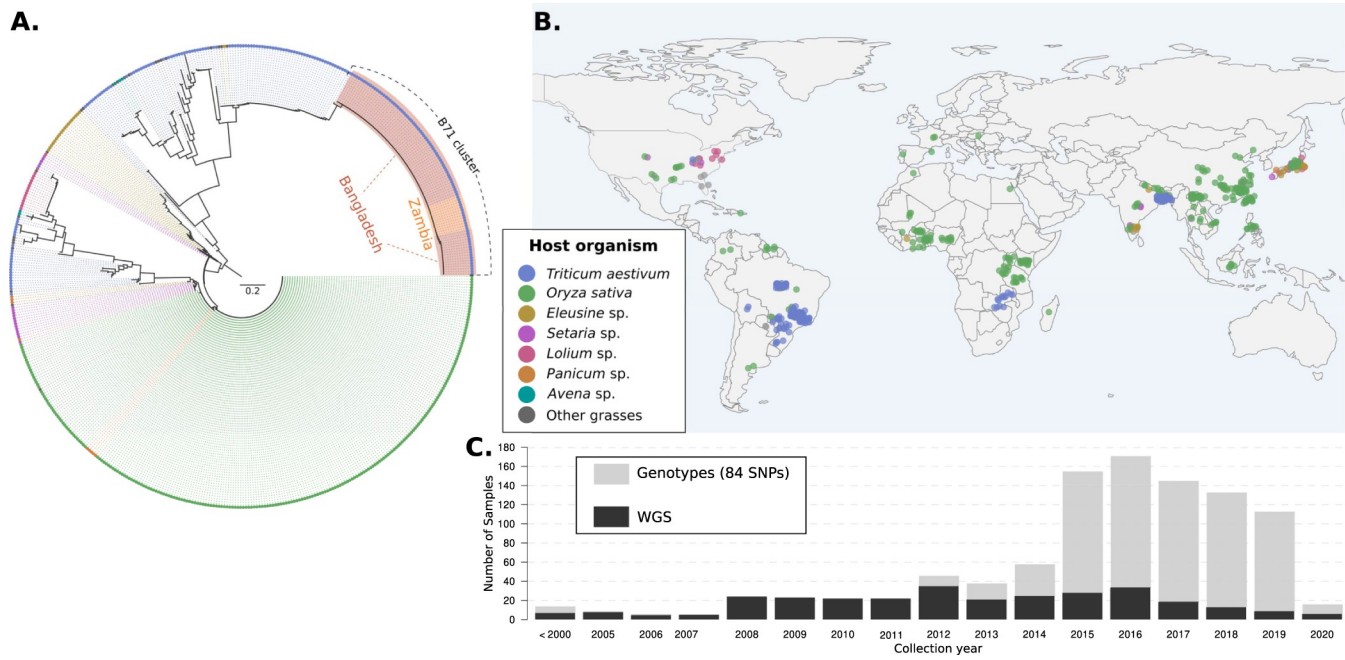

**Fig 1. The emergence of wheat blast in Bangladesh and Zambia was caused by the B71 genetic lineage of *Magnaporthe oryzae*. (A)** Neighbor-joining tree of 537 worldwide distributed *M. oryzae* isolates based on 84 concatenated SNPs obtained by multiplex amplicon sequence and/or genome sequences. The topology corresponds to the optimal tree drawn from 1,000 bootstrap replicates. Isolates that belong to the B71 lineage are shown with orange (13 Zambian isolates) and red (71 Bangladeshi isolates and the Bolivian B71) background shades. **(B)** Geographical distribution of *M. oryzae* samples used in the tree. The colored points represent the approximate geographical origin of the isolates. Colors in (A) and (B) correspond to the plant host organism (upper inset). The base map was created with the *R* package *rworldmap v.1.3.4 (data from: Natural Earth data v.1.4.0)*. **(C)** Distribution of the collection year of *M. oryzae* wheat-infecting isolates used in the phylogenetic analysis. The data underlying this figure can be found in https://doi.org/10.5281/zenodo.7590238. SNP, single nucleotide polymorphism.

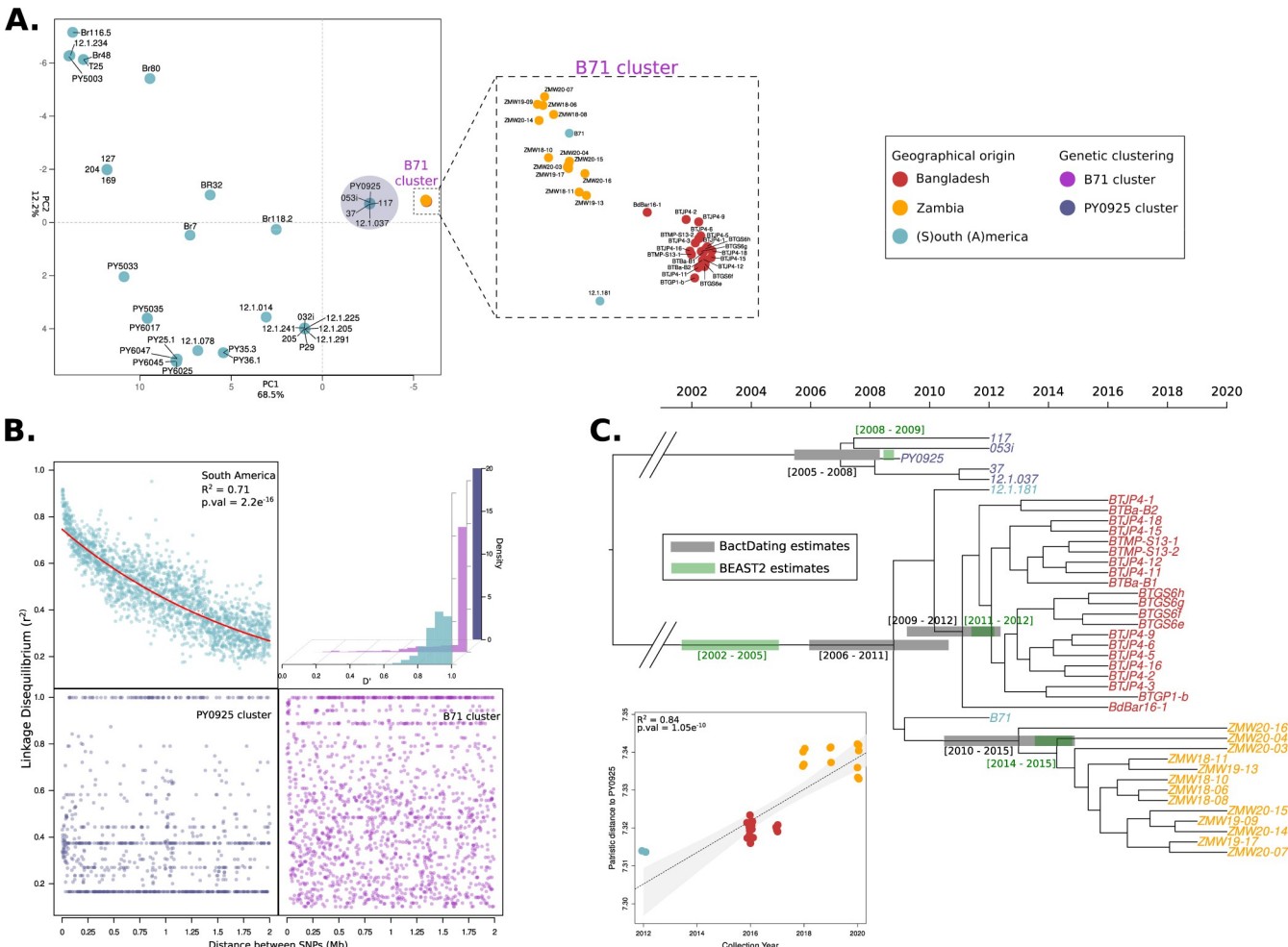

**Fig 2. Wheat blast outbreaks in Zambia and Bangladesh originated by independent introductions of the recently emerged B71 clonal lineage. (A)** The B71 lineage shows reduced genetic diversity in comparison with South American wheat-infecting isolates. PCA was performed based on genome-wide pairwise Hamming distances of 71 isolates from South America, Asia, and Africa. The colors of the points indicate the provenance of each isolate (see inset). The circular shaded area indicates isolates from the Brazilian cluster (PY0925) that is the closest to the B71 cluster. Axes labels indicate the percentage of total variation explained by each PC. The magnified area shows isolates that are part of the B71 lineage. **(B)** The B71 cluster is a clonal lineage. The scatter plots show pairwise LD (measured as $r^2$) between SNPs that are at most two megabases apart. The red solid line in the South American cluster represents a fitted exponential decay model using nonlinear least squares. The histograms display LD expressed as *D'* for genome-wide SNPs. The points and bars are colored as indicated in the inset. **(C)** The B71 clonal lineage has recently expanded with independent introductions in Zambia and Bangladesh. The scatter plot shows the linear regression (dotted line) of root-to-tip patristic distances (y-axis) versus collection dates (x-axis) for the isolates of the B71 clonal lineage. Maximum likelihood tip-calibrated time tree of the B71 cluster isolates (the PY0925 cluster was used as an outgroup). The horizontal gray and green bars indicate 95% CI for divergence dates (in calendar years) calculated using BacDating and BEAST2, respectively. The points and isolate names are colored as indicated in the inset. The data underlying this figure can be found in https://doi.org/10.5281/zenodo.7590238. CI, confidence interval; LD, linkage disequilibrium; PCA, principal component analysis; SNP, single nucleotide polymorphism.

most probably from South America, although we cannot totally rule out that the source population was located in an unsampled location outside of South America. These findings are consistent with the conclusions of a recent independent study [14].

We leveraged the collection dates of *M. oryzae* clonal isolates to estimate their evolutionary rate. Before performing the tip-calibration analyses [15], we removed regions that disrupt the clonal pattern of inheritance (S7–S9 Figs) [16] and tested for a correlation between genetic distances and collection years (S10 Fig). We obtained rates ranging from $2.74 \times 10^{-7}$ to $7.59 \times 10^{-7}$ substitutions/site/year (S3 Table), which were robust to the choice of both

substitution and clock models (S4 Table). Although these rates are approximately 9 times faster than previously calculated rates for the rice blast fungus [17–19], it is expected that evolutionary rates calculated from disease outbreaks, such as the cases in Bangladesh and Zambia, are likely faster due to incomplete purifying selection [20,21], and rates might vary in different blast fungus host-specific lineages. Using these rates, we dated the emergence of the Asian and African sub-lineage to similar periods (2009 to 2012 and 2010 to 2015, respectively) (Figs 2C and S11). The B71 clonal lineage itself dates back to a few years earlier and probably emerged in South America around 2002 to 2011, before spreading to other continents (Figs 2C and S11).

Can rapid genomic analyses inform practical disease management strategies of the wheat blast pandemic? Plant pathogens use secreted effector proteins to infect their hosts, but these effectors can also trigger plant immunity through an "avirulence" activity. The genome sequences of pandemic B71 lineage isolates offer the opportunity to identify effectors that can be targeted by the plant immune system. To date, the most predictive avirulence effector to confer wheat blast resistance is AVR-Rmg8 [22,23] that triggers an immune response in wheat carrying the resistance gene *Rmg8*. We scanned the available genomes and found that AVR-Rmg8 is conserved in all 36 isolates of the B71 clonal lineage even though the other 35 isolates of the *Triticum* lineage carry four diverse AVR-Rmg8 virulent alleles (eII, eII', eII", and eII"') that fully or partially evade immunity (Figs 3A and S12) [24]. B71 lineage isolates also lack the PWT4 effector, which is known to suppress AVR-Rmg8-elicited resistance [25].

These genome analyses predict that the B71 lineage isolates (AVR-Rmg8 positive, PWT4 negative) cannot infect wheat plants with the matching disease resistance gene *Rmg8*. To test this, we inoculated 14 B71 lineage isolates from Zambia and Bangladesh on wheat lines with and without the *Rmg8* resistance gene (Figs 3B and S13). Unlike a distinct South American isolate, none of these pandemic isolates could infect *Rmg8* wheat plants. We conclude that *Rmg8* is an effective resistance gene against the pandemic wheat blast lineage and has the potential to mitigate the spread of the disease.

Strobilurin fungicides are commonly used against the blast disease, but resistance is frequent in the South American population of the wheat blast fungus [26]. Genome analyses revealed that of the 71 wheat isolate genomes we examined, 13 carry the strobilurin resistance SNP (G1243C; Glycine to Alanine) in the mitochondrially encoded Cytochrome B (*CYTB*) gene (Fig 4A). Remarkably, all but one Brazilian isolate (12.1.181) of the 36 B71 lineage genomes carry the G1243C allele and are predicted to be strobilurin sensitive. We tested this by assaying B71 lineage isolates and found that all tested 30 isolates are strobilurin sensitive (Figs 4B and 4C and S14).

What is the evolutionary potential of the pandemic lineage of the wheat blast fungus? In laboratory experiments, we could readily recover spontaneous strobilurin (azoxystrobin)-resistant mutants of African isolate ZMW20-14 (Fig 4B and 4C) consistent with a high potential for emergence of fungicide resistance in the pandemic clonal lineage.

Multiple genetic lineages of the blast fungus are endemic to Africa causing destructive diseases on finger millet, rice, and other grasses [27,28]. The spread of the B71 clonal lineage to Africa raises the specter of sexual reproduction with endemic blast populations, which would further drive the evolutionary potential of the pandemic fungus. We determined that the pandemic B71 lineage belongs to the MAT1-2 mating type based on the genome sequences (S15 Fig) [29], and therefore, is predicted to mate with MAT1-1 isolates of *M. oryzae*. We tested and confirmed this prediction by showing that Zambian isolates from the pandemic lineage are fertile with MAT1-1 African finger millet isolates (Fig 4D and 4E and S5 Table).

The decade-old B71 clonal lineage of the wheat blast fungus, which spread twice from genetically diverse South American populations, happens to be avirulent on *Rmg8* wheat and

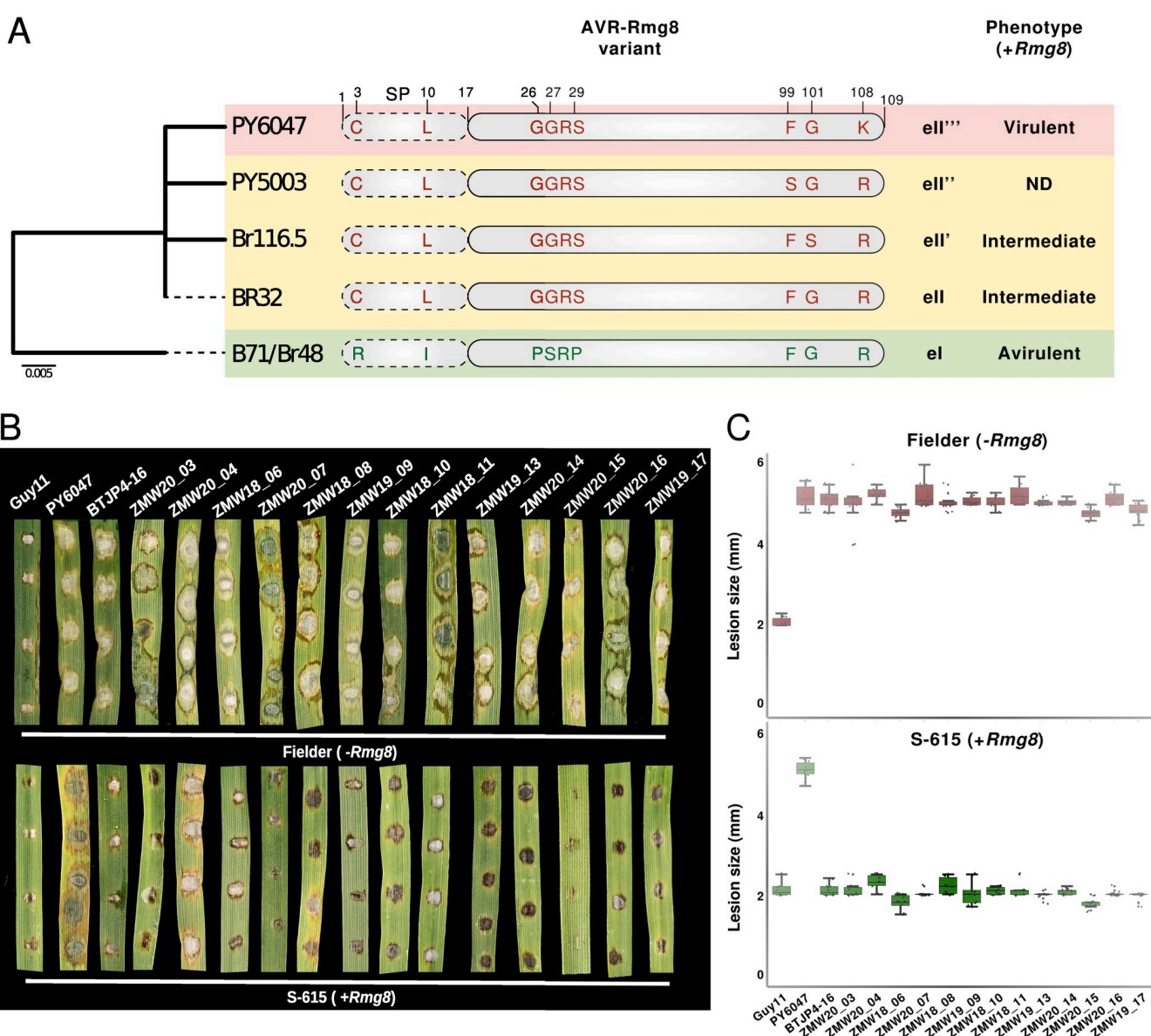

**Fig 3. Rmg8 confers resistance against Zambian wheat blast isolates. (A)** The wheat blast lineage contains 5 AVR-Rmg8 variants. Neighbor-joining tree based on amino acid sequences of all nonredundant AVR-Rmg8 variants of 71 wheat blast lineage isolates (left). Representative isolate IDs are shown for each branch. Schematic representation of polymorphic amino acids in AVR-Rmg8 variants of the wheat blast lineage (center). Virulence phenotype associated with each AVR-Rmg8 variant on *Rmg8* containing host plants (right). **(B)** The resistance gene *Rmg8* is effective against wheat blast isolates collected in Zambia. Leaves from two weeks old seedlings of Fielder (-*Rmg8*, upper panel) and S-165 (+*Rmg8*, lower panel) wheat cultivars were inoculated with spores from Zambian wheat blast isolates, the rice blast isolate Guy11 (non-adapted, avirulent control), PY6047 (virulent control; AVR-Rmg8 eII'''-carrier), and BTJP4-16 (avirulent on *Rmg8* carrying host plants, AVR-Rmg8 eI carrier). Disease and lesion size 5 days' post-infection. Photo taken by the authors. **(C)** Quantification of lesions size (in mm) of 10 leaves and three independent experiments. The data underlying this figure can be found in https://doi.org/10.5281/zenodo.7590238.

sensitive to strobilurin fungicides. However, the emergence of variants that are more damaging than the current genotypes is probable within short timescales. This could happen either through mutations or sexual recombination with endemic blast fungus populations. Such variants could have increased virulence and fungicide tolerance, thus adding to the difficulty in managing the wheat blast disease. These findings underscore the need for genomic

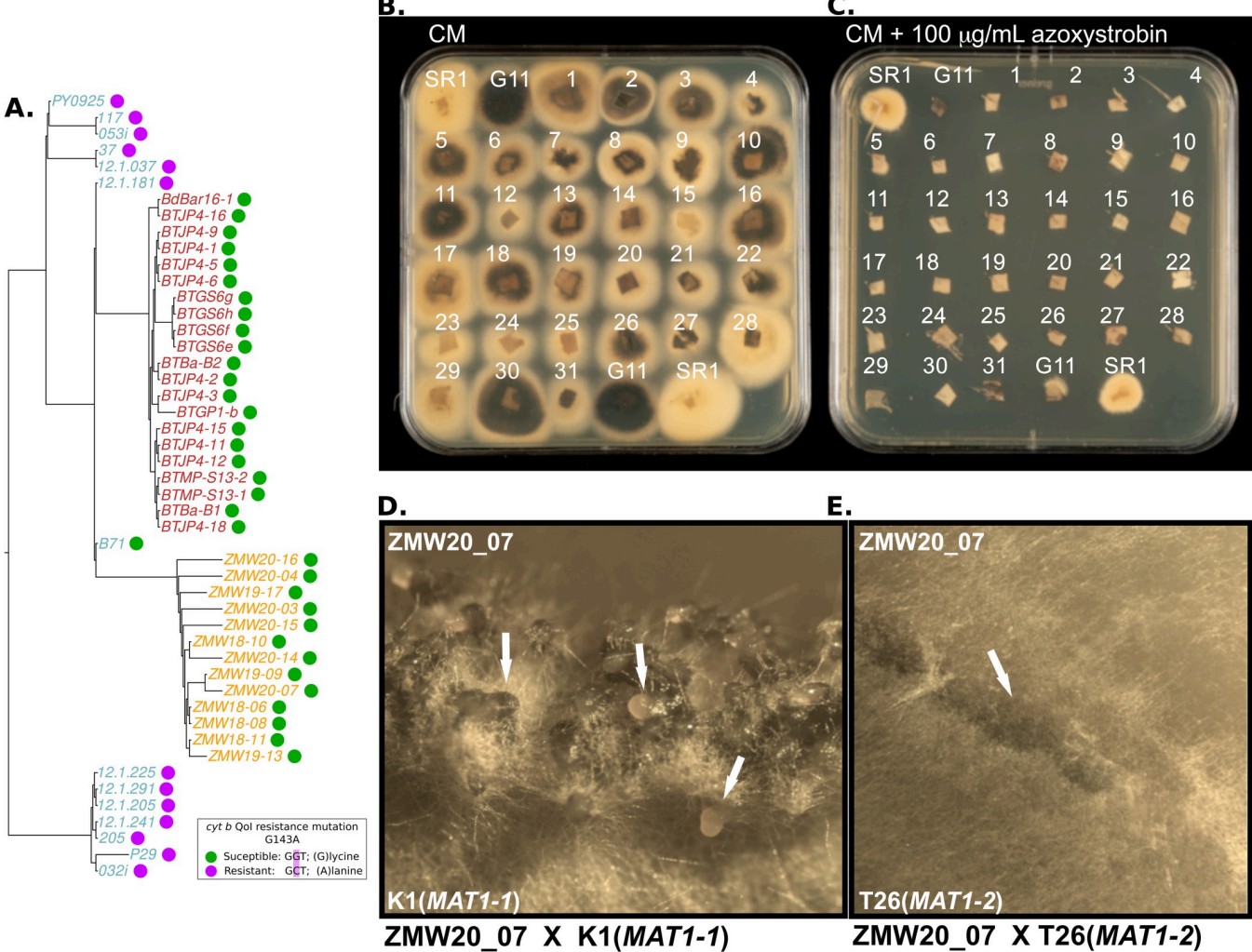

**Fig 4. Zambian wheat blast isolates are susceptible to strobilurin fungicides but at risk from resistance development and can mate with prevailing finger millet blast isolates.** (A) The tree describes, based on nuclear SNPs, the phylogenetic relationship among wheat-infecting blast isolates belonging to three clonal lineages: B71, PY0925, and P29. The tree was rooted in the midpoint. The colored dots next to each isolate label represent the resistant-type allele of the mitochondrially encoded *cyt b* gene associated with a susceptible or resistant predicted phenotype as shown in the inset. **(B, C)** $4 \times 10^6$ spores of the Zambian strain ZMW20-14 were exposed to 100 µg ml[1] azoxystrobin to obtain an azoxystrobin-resistant strain with the G143A mutation (SR1). This strain, the rice infecting strain Guy11 (G11), and the wheat-infecting strains numbered 1–31, respectively, BTBa-B1, BTBA-B2, BTGS6e, BTGS6f, BTGS6g, BTGS6h, BTGP1-b, BTMP-S13-1, PY6047, BTJP4-1, BTJP4-2, BTJP4-3, BTJP4-5, BTJP4-6, BTJP4-9, BTJP4-11, BTJP4-12, BTJP4-15, BTJP4-16, BTJP4-18, ZMW18-06, ZMW20-07, ZMW18-08, ZMW19-09, ZMW18-10, ZMW18-11, ZMW19-13, ZMW20-14, ZMW20-15, ZMW20-16, and ZMW19-17 were grown for 4 days at 25°C on CM in the absence (A) and presence (B) of azoxystrobin indicating that all the Zambian and Bangladeshi isolates have the "strobilurin susceptible" genotype as anticipated by their *CYTB* sequences. **(D)** Zambian isolate ZMW20-7 (MAT-1-2) successfully produced perithecia when crossed with a finger millet blast isolate K1 (MAT-1-1) but **(E)** ZMW20-7 was unable to produce perithecia when crossed with a finger millet blast isolate T26 of the same mating type (MAT-1-2). Photos taken by the authors. The data underlying this figure can be found in https://doi.org/10.5281/zenodo.7590238. SNP, single nucleotide polymorphism.

surveillance to improve tracking and monitoring of the wheat blast fungus on a global scale and identifying variants of concern as soon as they emerge [3].

Deployment of *Rmg8* wheat varieties on their own in affected areas is unlikely to provide sustained wheat blast management given that the fungus will probably evolve virulent races through AVR-Rmg8 loss of function mutations or gain of PWT4, an effector that suppresses the resistance conferred by Rmg8 [25]. *Rmg8* should therefore be combined with other sources of resistance as previously shown [24,30] to be fully useful in areas with high fungal load and

disease pressure. However, it might be judicious in the short term to breed and deploy Rmg8 varieties in high-risk areas such as regions neighboring affected countries. This could serve as a firebreak to alleviate the need for wheat cultivation holidays as implemented in the Indian province of West Bengal, particularly in the face of likely wheat shortages and disruptions to global wheat trade [8,31].

## Materials and methods

### Selection of SNP panel for multiplex amplicon sequencing

To identify the SNPs that could be used for genotyping of *M. oryzae* from Bangladesh by multiplex amplicon sequencing (Floodlight Genomics, https://floodlightgenomics.com), we filtered 15,871 SNPs identified in Islam and colleagues (2016) [10] using the following criteria: (i) they had to be polymorphic among wheat blast strains alone to make the markers useful for diagnostics; (ii) the minor allele was >30%; (iii) SNPs were located on long exonic sequences (>1,500 bp without interrupting intron); and (iv) long exons to contain only 2 to 4 SNPs. The last two criteria were to make sure that the assay will focus on SNPs surrounded by well-conserved stretches among wheat blast strains with an aim to reduce amplification failures due to polymorphism in the primer binding sites. The above criteria reduced the available genomic regions to 102 loci. We designed 102 PCR primer pairs to amplify approximately 200 bp amplicon for each gene containing 100 bp flanking regions on each side of the SNP locus for multiplex amplicon sequencing. Trial sequencing runs with these primer pairs resulted in 84 primer pairs that consistently produced amplicons that can be sequenced to identify the correct nucleotides in those SNP loci in Bangladeshi wheat blast isolates [32]. We used these as a panel of 84 SNPs to discriminate between the wheat blast clonal lineage of *M. oryzae* in Bangladesh from other genotypes. Initial analyses and the use of the 84 SNP panel was reported in [32], and the details of 84 SNPs including gene names and primer sequences along with the dataset for benchmarking are available in Tembo and colleagues (2021) [33].

### Phylogenetic placement of *Magnaporthe oryzae* wheat-infecting isolates from the Zambian wheat blast outbreak (2018 to 2020) using a set 84 SNPs

To establish the genetic makeup and the phylogenetic placement of the wheat-infecting lineage that caused a wheat blast outbreak in Zambia (2018 to 2020), we analyzed a set of 84 SNPs, which were designed to distinguish between the pandemic clonal lineage of the wheat blast fungus (*Magnaporthe oryzae*) that reached Southeast Asia in 2016 from other *M. oryzae* genotypes [10]. The dataset included 237 *M. oryzae* isolates from 13 different grasses that were genotyped using multiplex amplicon sequencing ("MonsterPlex") [33,34]. MonsterPlex is a highly accurate and cost-effective genotyping method to rapidly genotype field isolates [32,33]. The Monsterplex genotypes were extracted from the genotyping matrices [33,34] as previously described [35]. To complement the MonsterPlex dataset and increase the geographic breadth of *M. oryzae* isolates, we extracted the 84 SNPs from 351 publicly available *M. oryzae* genomes. The joint dataset consisted of 537 worldwide distributed *M. oryzae* isolates (Figs 1 and S1 and S1 Table).

After removing adapters with *AdapterRemoval* v2 [36], we mapped the Illumina-derived short reads to the *M. oryzae* B71 reference genome [37] using *BWA mem* [38]. To identify the genomic location of the 84 diagnostic SNPs in the *M. oryzae* B71 reference genome, we aligned each diagnostic SNP surrounded by 100 bp surrounding regions to the B71 reference genome using *blastn* [39]. Afterwards, for each of the per isolate mapped BAM files, we retrieved the

genotypes at each of the 84 genomic locations using *samtools mpileup* [40]. We concatenated all SNPs in a multi-fasta-like file that was used as input for phylogenetic analyses.

We built a Neighbor-Joining tree that includes a total of 537 *M. oryzae* isolates using the *R* package *phangorn* [41]. We displayed a tree topology that corresponds to the optimal tree drawn from 1,000 bootstrap replicates (Figs 1 and S1).

We further estimated the accuracy of the genotyping method by comparing SNP data acquired from 51 isolates using MonsterPlex to the SNPs extracted from matching genome sequences. We found that SNPs from both acquisition methods were identical whenever a SNP call was made (100%, 4,251/4,251 sites). In total, only 33 sites had gaps with missing data from MonsterPlex (0.77%, 33/4,284 sites) (S1 Table).

To show that the set of 84 SNPs are informative, we compared the genetic (Hamming) distances between each pair of blast isolates using the set of 84 SNPs and the genome-wide SNPs. Our analysis revealed a correlation coefficient of 0.82 between the two sets of pairwise distances (S2C Fig). To show that this correlation is robust and much higher than expected by chance, we repeated the calculation of pairwise distances for both datasets (84 SNPs and genome-wide SNPs) randomly subsampling a subset of the isolates (10%) with and without isolate names permutation with 100 repetitions. This analysis revealed a median correlation of pairwise distances of 0.82 and 0.001 for the resampling with permuted distances and sample pairs (S2C Fig). We repeated the analysis using only pairwise distances among wheat-infecting isolates and obtained a correlation coefficient of 0.9 (S2D Fig), which shows that the set of 84 SNPs accurately reflect the genetic diversity of the wheat blast fungus.

## Processing of short reads and variant calling

Our phylogenetic analyses based on 84 SNPs (Figs 1 and S1) confirmed our previous analyses, which showed that the emergence of wheat blast in Zambia and Bangladesh was caused by the same pandemic lineage of *M. oryzae* that traces its origin in South America [32,35,42]. Consequently, from here on, we analyzed a set of 71 wheat-infecting *M. oryzae* genomes that belong to the wheat-infecting genetic lineage. This set included isolates from South America ($N = 37$), the outbreak in Bangladesh (2016 to 2018) ($N = 21$), and isolates from the outbreak in Zambia (2018 to 2020) ($N = 13$) that were exclusively sequenced for this study [43,44] (S2 Table).

We removed adapters from the short reads from the set of 71 *M. oryzae* genomes using *AdapterRemoval* v2 [36]. To avoid reference bias, a common problem in population genetic analysis when comparing genetically similar populations, we mapped the short reads to the rice-infecting 70–15 reference genome [45] using *BWA mem2* [46,47] (S2 Table). We used *samtools* [40] to discard non-mapped reads and *sambamba* [48] to sort and mark PCR optical duplicates.

As a first step to call variants, we used *HaplotypeCaller* from *GATK* [49,50] to generate genomic haplotype calls per individual using the duplicate-marked BAM files as input. Subsequently, we used *CombineGVCFs*, *GenotypeGVCFs*, and *SelectVariants* from *GATK* [49] to combine the individual genomic VCFs, call genotypes, and filter SNPs, respectively. In order to select high-quality SNPs to be included in our phylogenetic and population genetic analyses, we filtered SNPs using *Quality-by-Depth* (*QD*), which is one of the per-SNP summary statistics generated by GATK. Using *M. oryzae* genomes sequenced by two different sequencing technologies, we have previously shown that QD is the summary statistic that generates the best true positive and true negative rates [19]. Following this approach, using *GATK VariantFiltration [49]*, we kept SNPs with QD values no further away than one standard deviation unit away from the median of the distribution of QD values [19]. Finally, we created a new VCF file keeping only non-missing positions using *bcftools* [40].

## Population structure analyses

To assess the population structure of the 71 *M. oryzae* isolates, we used genetic distances coupled with dimensionality reduction methods. First, we calculated pairwise Hamming distances using *Plink V.1.9* [51]. These distances were used as input for PCA using the function *prcomp* from the *R* package *stats* [52] (Fig 3A). Additionally, we used f3-outgroup statistics [53] to establish the pairwise relatedness between *M. oryzae* samples (X and Y) after divergence from an outgroup: *f3*(X, Y; outgroup). We used the rice-infecting *M. oryzae* 70–15 as an outgroup [45]. We calculated *z*-scores for every possible pairwise sample comparison included in the f3-statistics test (*N* = 2,485). Subsequently, we carried out hierarchical clustering using the function *hclust* from the *R* package *stats* [52]. As input, we used a distance matrix generated from the f3-statistics-derived *f3* values (S3 Fig).

## Distinguishing clonality from outcrossing

To distinguish clonality from outcrossing in the B71 pandemic lineage and other genetic groups identified in our population structure analyses, we used patterns of LD decay. While sexual reproduction (outcrossing) will generate patterns of LD decay that are driven by meiotic recombination, LD is not expected to decay in asexual non-recombining populations, i.e., the whole genome will be in LD. We analyzed LD decay patterns in the B71 lineage, the PY0925 lineage and treated the rest of Brazilians *M. oryzae* isolates as one group. To compute standard measures of LD, we used *VCFtools* [54]. Since *VCFtools* is designed to handle diploid organisms, we transformed the haploid *VCF* files into "phased double haploid" *VCF* files. Afterwards, we computed pairwise LD as $r^2$ and Lewontin's *D* and *D'*. To evaluate LD decay, for each of the genetic groups, we calculated the average of each LD measure ($r^2$, Lewontin's *D* and *D'*) in bins of physical genomic distance (Figs 2B and S4). To quantify the significance of LD decay, we fitted an exponential decay model using nonlinear least squares. In order to compare the patterns of LD decay between the clonal lineages and the Brazilian group, we downsample the number of SNPs in the Brazilian group to the number of SNPs segregating in the B71 clonal lineage.

To perform a qualitative comparison between the observed patterns of LD decay with the expectations of LD decay in idealized populations, we performed forward simulations using *FFPopSim* [55]. We simulated genomes that consisted of 300 equidistant SNPs. We first sought to ascertain the effect of the probability of sexual reproduction per generation on the patterns of linkage disequilibrium decay. Each simulation was carried out for 100 generations keeping the population size, the crossover probability, and the mutation rate constant, but changing the probability of sexual reproduction per generation (0, $10^{-3}$, $10^{-2}$, $10^{-1}$, and 1) (S5 Fig). Additionally, we investigated the effect of the population size on the patterns of LD decay. To this purpose, we simulated genomes that consisted of 200 equidistant SNPs. Each simulation was carried out for 100 generations keeping the crossover probability, the mutation rate, and the probability of sexual reproduction per generation constant, but changing the population size parameter ($10^2$, $10^3$, $10^4$, $10^5$) (S6 Fig).

Finally, to detect recombination events in each of the genetic groups, we performed the four-gamete test [56] using *RminCutter* [57]. To be able to compare the number of violations of the four-gamete test among genetic groups, we normalized the number of violations of the four-gamete test by the number of segregating SNPs per genetic group (S7 Fig).

## Phylogenetic analyses, estimation of evolutionary rates, and divergence times

To carry out phylogenetic analyses, we used only the non-recombining genetic groups (clonal lineages) B71 and PY0925 (the latter was used as an outgroup) and included exclusively positions with no-missing data (full information). First, we generated a maximum likelihood (ML)

phylogeny using *RAxML-NG* with a GTR+G substitution model and 1,000 bootstrap replicates [58]. Then, we used the best ML tree as input for *ClonalFrameML* [16], a software that detects putative recombination events and takes those into account in subsequent phylogenetic-based inferences. For every isolate, we calculate the percentage of total SNPs masked by *ClonalFrameML* (S8 Fig). Additionally, we used pairwise Hamming distances to evaluate the levels of intra- and inter-outbreak genetic variation before and after *ClonalFrameML* filtering (S9 Fig).

Since the LD decay analyses revealed that the B71 pandemic lineage is a non-recombining clonal lineage, we hypothesized that the SNPs marked as putatively affected by recombination are preferentially located in genomic regions affected by structural variants, e.g., presence/absence variants. Such variants will generate phylogenetic discordances due to differential reference bias among the B71 isolates. To test this hypothesis, we created full-genome alignments of the B71 and the 70–15 reference genomes using *Minimap2* [59] and visualized the output with *AliTV* [60]. Then, we overlapped the visual output with the SNPs putatively affected by recombination that were previously identified by *ClonalFrameML* (S10 Fig).

To test for the existence of a phylogenetic temporal signal (i.e., a positive correlation between sampling dates and genetic divergence), we used the recombination-corrected tree generated by *ClonalFrameML* (Fig 2C). Using this tree, and without constraining the terminal branch lengths to their sampling times, we measured tip-to-root patristic distances between all isolates of the B71 clonal lineage and the outgroup clonal lineage PY0925 with the *R* package *ape* [61]. We calculated the Pearson's correlation coefficient between root-to-tip distances and sampling dates, and fitted a linear model using them as response and linear predictor, respectively (*distances ~ sampling dates*) (Fig 2C). We calculated 95% confidence intervals of the correlation signal between root-to-tip distance and collection dates by sampling with replacement and recalculating the correlation coefficient 1,000 times (Figs 2C and S11). Additionally, to demonstrate that the obtained correlation coefficient was higher than expected by chance, we performed 1,000 permutation tests, where the collection dates were randomly assigned to the wheat blast isolates (S11 Fig).

To estimate the evolutionary rate and generate a dated phylogeny, where the divergence dates of all common ancestors are estimated, we used two different approaches. First, we used *BactDating*, a Bayesian framework that estimates evolutionary rates and divergence times using as an input a precomputed phylogenetic reconstruction [62]. As input for *BactDating*, we used the recombination-corrected tree generated by *ClonalFrameML*. The tree was loaded into *BactDating* using the function *loadCFML*, which permits the direct use of the output of *ClonalFrameML* as input for *BactDating* without the need of correcting for invariant sites (Fig 2C). In our second approach, we used *BEAST2*, a Bayesian approach, which in contrast to BactDating, co-estimates the evolutionary rate and the phylogenetic reconstruction [63]. From the alignment of the concatenated SNPs, we masked those that *ClonalFramML* marked as putatively recombining and used the masked alignment as input for the BEAST2 analyses. We carried out a Bayesian tip-dated phylogenetic analyses using the isolates' collection dates as calibration points. Since, for low sequence divergence (<10%), different substitution models tend to generate very similar sequence distance estimates [64], to simplify the calculation of parameters, we used the HKY substitution model instead of more complex models such as GTR. To reduce the effect of demographic history assumptions, and to calculate the dynamics of the population size through time, we selected a Coalescent Extended Bayesian Skyline approach [65]. The evolutionary rate was co-estimated using a strict clock model and a wide uniform distribution as prior (1E-10 - 1E-3 substitutions/site/year), which permits a broad exploration of the MCMC chains. To test the robustness of our evolutionary rate estimation to changes in substitution and clock models, we repeated the analysis using GTR in combination with a strict clock model and HYK in combination with a random local clock model [66]. To have

genome-scaled results, the invariant sites were explicitly consider in the model by adding a "*constantSiteWeigths*" tag in the XML configuration file. Using the CIPRES Science Gateway [67], we ran four independent MCMC chains, each of which had a length of 20,000,000 with logs every 1,000 iterations. We combined the outputs with the *LogCombiner* and used *TreeAnotator* [63,68] to calculate the Maximum Credibility Tree as well as dating and support values for each node (Figs 2C and S12 and S3 and S4 Tables).

## Determination of mating types

To assign the mating type for each isolate, we used two approaches. First, we created a fasta file containing the nucleotides codifying for the two mating type loci: MAT1-1-1 (GenBank: BAC65091.1) and MAT1-2-1 (GenBank: BAC65094.1) and used it as a reference genome. Using *bwa-mem2* [46,47], we mapped each isolate and used *samtools depth* [40] to calculate the breadth of coverage for each locus as a proxy for the mating type assignment.

## Identification of AVR-Rmg8 effector variants and generation of the Avr-Rmg8 family tree

We used a mapping approach to identify Avr-Rmg8 family members in all 71 wheat blast lineage genomes. We mapped short reads sequencing data of each isolate to the B71 reference genome assembly [37] using *BWA mem2* [38] and extracted the consensus sequence of the AVR-Rmg8 locus from the output alignment files using *SAMtools* v.1.9 and *BCFtools* v.1.9 [40] for each isolate. This led to the identification of five AVR-Rmg8 variants in 71 sequences.

## Generation of the Avr-Rmg8 family tree

To infer the AVR-Rmg8 family tree, we generated a multiple sequence alignment of 108 amino acid sequences of all identified variants using *MUSCLE* [69]. We then removed pseudogenes and duplicated sequences to generate a nonredundant dataset and generated a maximum-likelihood tree with 1,000 bootstrap replications in *MEGA7* [70].

## Leaf-drop and spray infection assay

To evaluate the response of Rmg8 against wheat blast isolates from Zambia, we carried out leaf drop and spray inoculations. An *Oryza*-infecting isolate from French Guyana, Guy11 was used as a negative control as this isolate does not infect wheat, while a *Triticum*-infecting isolate PY6047 from Brazil was used as a positive control it lacks the virulent allele of *AVR-Rmg8* [71]. Another *Triticum*-infecting Bangladesh isolate BTJP4-16 that carries an avirulent allele of *AVR-Rmg8* was also included. Wheat cultivars, S-615 (+Rmg8) and fielder (-Rmg8) were grown for 14 days in 9-cm diameter plastic plant pots or seed trays. Detached leaves from two weeks old seedlings were inoculated with fungal conidial suspension diluted to a final concentration of $1 \times 10^5$ conidia mL-1 using drop inoculation method. The inoculated detached leaves were incubated in a growth room at 24°C with a 12 h light period. Alternatively, plants were inoculated by spray infection using an artist's airbrush (Badger, United States of America) as previously described in [72]. After spray inoculation, the plants were covered in polythene bags and incubated in a negative pressure glasshouse with a 12 h light and dark cycle. Disease severity was scored after 5 to 6 days by evaluating lesion color and count or color and size for spray infection or drop inoculation, respectively. Lesions were classified as small black/brown non-spreading spots for resistant and large gray-spreading for susceptible. Each infection experiment was carried out three times.

## Generation of crosses and fertility status analysis

Generation of genetic crosses was carried as previously described in [73]. A subset of Zambian isolates ZMW20_04, ZMW18_06, ZMW20_07, ZMW18_10, ZMW20_15, and ZMW20_16 (*MAT-1-2*) were tested against two finger millet tester isolates from Tanzania, T15 (*MAT-1-1*) or T26 (*MAT-1-2*), one from Kenya K1(*MAT-1-1*), and one from Ethiopia E12 (*MAT-1-1*). Two isolates were cultured opposite each other on oatmeal agar plate and incubated at 24˚C for 7 to 10 days. The cultures were then transferred to a 20˚C incubator until flask-shaped perithecia appeared at the crossing point. A Leica DFC360 FX microscope (Leica, Wetzlar, Germany) was used to visualize and image the formation of perithecia.

## Isolation of azoxystrobin-resistant *Magnaporthe oryzae* strains

Isolation of azoxystrobin-resistant *Magnaporthe* strains was carried out by exposure of spores of the fungus to azoxystrobin at 100 g ml$^{-1}$. Four million conidia of the Zambian wheat-infecting isolate ZMW20-14 were recovered from CM grown agar cultures and plated on the surface of six 12 cm$^2$ square Minimal medium agar plates (MM; [74]) with azoxystrobin at 100 g ml$^{-1}$. Plates were incubated for 2 weeks and then overlaid with MM plus 100 μg ml$^{-1}$ azoxystrobin and grown for a further week. Five colonies were isolated from these plates and colonies were purified by subculture to MM with 100 μg ml$^{-1}$ azoxystrobin for a further week before genotyping. PCR competent genomic DNA was isolated from a 4 mm$^2$ plug of mycelium from the purified azoxystrobin-resistant colonies (named AZ1-AZ5) with disruption using an automated tissue homogenizer and cell lyser. To extract the mycelial plug, it was placed in 400 l of extraction buffer (1 M KCl, 10 mM Tris-HCl (pH 8), 5 mM EDTA (pH 8) together with a 5 mm stainless steel bead (Qiagen)) and macerated at 1,200 rpm for 2 min in the Geno/grinder followed by a 10-min centrifugation (17,000 x g) in a bench top centrifuge at room temperature, and 300 l of the supernatant was transferred to a new tube and nucleic acids were precipitated with 300 l of isopropanol followed by a 10-min centrifugation at room temperature at 17,000 x g. Nucleic acid pellets were air dried for 10 min and then resuspended in 50 μl of TE with 50 μg ml$^{-1}$ RNAse by vortexing and then incubated for 10 min at 37˚C, and 1 l of the DNA was used in a 50 l PCR reaction with the enzyme Q5 polymerase (New England Biolabs) and the primers Cytb-f AGTCCTAGTGTAATGGAAGC and Cytb-r ATCTTCAACGTGTT TAGCACC (annealing temperature 61.5˚C). Amplicons were sequenced following gel purification in both directions using the same primers used to generate them to score for the presence or absence of the strobilurin resistance conferring mutation [26].

## Supporting information

**S1 Fig. The emergence of wheat blast in Bangladesh and Zambia was caused by the B71 genetic lineage of *Magnaporthe oryzae*.** Neighbor-joining tree of 576 worldwide distributed blast isolates based on 84 concatenated SNPs. The topology corresponds to the optimal tree drawn from 1,000 bootstrap replicates. Names of host organisms are shown together at the tips. Duplicated identifiers without the "_*g*" suffix, represent data belonging from the same isolate but from the "monsterplex" set of 84 SNPs. The data underlying this figure can be found in https://doi.org/10.5281/zenodo.7590238.
(EPS)

**S2 Fig. The set of 84 Monsterplex SNPs reflects the patterns of genome-wide diversity of the blast fungus.** Neighbor-joining tree of 284 worldwide distributed *M. oryzae* isolates based on 84 concatenated SNPs **(A)** or genome-wide SNPs **(B)**. **(C)** The scatter plot shows genetic distances between each pair of blast isolates using the set of 84 SNPs and the genome-wide

SNPs. The boxplots show the correlations of genetic distances between each pair of isolates using the set of 84 SNPs and the genome-wide SNPs. The distributions were generated by randomly subsampling a subset of the isolates (10%) with and without isolate names permutation 100 times. **(D)** The scatter plot shows pairwise genetic distances including only the wheat-infecting blast isolates for the set of 84 SNPs and the genome-wide SNPs. The data underlying this figure can be found in https://doi.org/10.5281/zenodo.7590238.
(EPS)

**S3 Fig. Genetic clustering of *Magnaporthe oryzae* identifies isolates from the Bangladesh and Zambian outbreaks as part of the B71 lineage.** The pairwise relatedness between *M. oryzae* samples (X and Y) was estimated using *f3*-outgroup statistics of the form *f3*(X, Y; outgroup), which measures the amount of shared genetic history (genetic drift) between X and Y after the divergence from an outgroup (rice-infecting *M. oryzae* isolate 70–15). The hierarchical clustering is based on *f3*-scores resulting from *f3*-outgroup statistic calculations. Darker colors indicate more shared drift. The data underlying this figure can be found in https://doi.org/10.5281/zenodo.7590238.
(EPS)

**S4 Fig. Linkage disequilibrium (LD) does not decay with physical distance in wheat-infecting blast isolates from the B71 lineage.** The scatter plots show pairwise LD (measured as *D*) between SNPs that are at most two megabases apart. The solid lines represent fitted linear and exponential decay models as indicated in the inset. The data underlying this figure can be found in https://doi.org/10.5281/zenodo.7590238.
(EPS)

**S5 Fig. Forward simulations indicate that the probability of sexual reproduction per generation determines the extent of LD decay.** The simulated genomes consisted of 300 equidistant SNPs. Each simulation was carried out for 100 generations keeping the population size, crossover probability, and the mutation rate constant, but changing the probability of sexual reproduction per generation (see inset). Dots represent LD (measured as *D*) as a function of the distance between two loci and thick lines represent the mean value per distance-bin. Points and lines are colored as indicated in the inset. The data underlying this figure can be found in https://doi.org/10.5281/zenodo.7590238.
(EPS)

**S6 Fig. Forward simulations indicate that LD breaks as a function of population size.** The simulated genomes consisted of 200 equidistant SNPs. Each simulation was carried out for 100 generations keeping the crossover probability, the mutation rate, and the probability of sexual reproduction per generation constant, but changing the population size parameter. Dots represent LD (measured as *D*) as a function of the distance between two loci. The data underlying this figure can be found in https://doi.org/10.5281/zenodo.7590238.
(EPS)

**S7 Fig. *ClonalFrameML* identifies on average 16% of SNPs/per isolate as putatively recombining.** The dendrogram shows the phylogenetic relationships of *Magnaporthe oryzae* strains as inferred by *RAxML-NG*. The dendrogram is schematic, i.e., the branch lengths do not accurately represent phylogenetic distances. The bars show the percentage of SNPs identified as putatively recombining by *ClonalFrameML*, which were masked in our dating analyses. The bars and isolate names are colored as indicated in the inset. The data underlying this figure can be found in https://doi.org/10.5281/zenodo.7590238.
(EPS)

**S8 Fig. The outbreaks of Bangladesh and Zambia show similar levels of genetic diversity.** The panels show the total number of segregating SNPs in the outbreaks of Zambia, Bangladesh, and the B71 clonal lineage. **(A)** Total number of segregating SNPs. **(B)** Total number of SNPs after excluding putatively recombining SNPs identified *ClonalFrameML*. All groups include 13 isolates that were sampled with replacement 100 times. The data underlying this figure can be found in https://doi.org/10.5281/zenodo.7590238.
(EPS)

**S9 Fig. Putative recombinant regions are likely caused by structural variation. (A)** The upper horizontal track is a representation of the wheat blast B71 reference genome. The bottom horizontal track is a representation of the 70–15 rice blast reference genome. Vertical ticks represent different types of SNPs (dark blue: unmasked SNPs; light blue: partially masked SNPs, i.e., masked in only some samples; and red: SNPs masked in all samples) (inset). Unmasked and partially masked SNPs were included in the phylogenetic analyses, whereas fully masked SNP were excluded from them. Ribbons represent syntenic regions between the rice blast 70–15 genome and the wheat blast B71 reference genomes. The coloring of the ribbons indicates the level of identity (chromatic scale). **(B)** The bar plot summarizes the proportion of the three SNP categories (inset) per bin of link identity between the B71 and 70–15 reference genomes. The data underlying this figure can be found in https://doi.org/10.5281/zenodo.7590238.
(EPS)

**S10 Fig. The temporal signal of the B71 lineage is robust and significantly bigger than expected by chance.** Boxplots represent the distribution of the Pearson's *r* correlation between root-to-tip patristic distances and collection dates of the isolates of the B71 clonal lineage. The left boxplot depicts the distribution of 1,000 instances of sampling with replacement from the original dataset. The right boxplot represents the distribution of 1,000 permutation tests, where collection dates were randomly assigned to wheat blast isolates. Median values are shown within each boxplot. The data underlying this figure can be found in https://doi.org/10.5281/zenodo.7590238.
(EPS)

**S11 Fig. Bayesian tip calibrated phylogenetic tree using individuals belonging to B71 and PY0925 clonal lineages.** Average and HPD 95% confidence intervals are shown in calendar years for nodes leading to the outbreaks or clonal lineage expansions. The data underlying this figure can be found in https://doi.org/10.5281/zenodo.7590238.
(EPS)

**S12 Fig. The pandemic wheat blast lineage contains the avirulent AVR-Rmg8 variant eI.** Neighbor-joining tree based on amino acid sequences of all AVR-Rmg8 variants of the wheat blast lineage reveals diversifying AVR-Rmg8 alleles. Isolate IDs are shown for each branch. Wheat blast isolates of the pandemic B71 lineage are colored. The data underlying this figure can be found in https://doi.org/10.5281/zenodo.7590238.
(EPS)

**S13 Fig. Rmg8 confers resistance against the Zambian wheat blast population. (A)** Leaves from two weeks old seedlings of Fielder (-Rmg8) and **(B)** S-165 (+Rmg8) wheat cultivars were inoculated with spores from ZMW20_07, ZMW18_10, ZMW20_16, Guy11, PY6047, and BTJP4-16 using a spray infection method. Disease lesions were scored six days' post-infection. Photo taken by the authors. **(C, D)** Boxplots show lesion count per 10 cm for two independent experiments. The data underlying this figure can be found in https://doi.org/10.5281/zenodo.

7590238.
(EPS)

**S14 Fig. All Zambian and Bangladeshi wheat-infecting blast isolates are susceptible to strobilurin fungicides. (A)** The 70 wheat blast isolates had just two genotypes with respect to the mitochondrially encoded gene *CYTB*. The GGT to GCT mutation in the *CYTB* gene results in a substitution at position 143 in the gene product and is known to confer resistance to strobilurin class fungicides. All the Zambian isolates sequenced have the "strobilurin susceptible" genotype. **(B)** Sequencing of the *CYTB* partial gene sequence in the azoxystrobin-resistant strain (SR1) indicated a homogenous population of mitochondria with the CytB G143A genotype. The data underlying this figure can be found in https://doi.org/10.5281/zenodo.7590238.
(EPS)

**S15 Fig. Only the MAT 1–2 segregates in the B71 clonal lineage.** The Neighbor-Joining tree describes the phylogenetic relations among wheat-blast–infecting isolates. The mating types are codified as colored circles next to the isolate name (see inset). The data underlying this figure can be found in https://doi.org/10.5281/zenodo.7590238.
(EPS)

**S1 Table. Dataset of isolates used in the Monsterplex Analysis.** Isolates with both types of data were used for Quality Control analyses.
(TSV)

**S2 Table. Dataset of selected wheat-infecting isolates with genome-wide information.**
(TSV)

**S3 Table. Estimated evolutionary rates for wheat-infecting *Magnaporthe oryzae* lineages measured as substitutions/site/year.**
(TSV)

**S4 Table. Estimated evolutionary rates using BEAST2 for wheat-infecting *Magnaporthe oryzae* lineages measured as substitutions/site/year, under different substitution and clock models.**
(TSV)

**S5 Table. Mating type crosses and fertility status.**
(TSV)

## Acknowledgments

We thank Aida Andrés, members of her group at UCL and Michael Dannemann for input on data analyses, and Talia Karasov for comments on the manuscript. We also thank Emilie Chanclud, as well as Emerson M. Del Ponte and group for contributions to the genotyping experiments.

## Author Contributions

**Conceptualization:** Sergio M. Latorre, Vincent M. Were, Andrew J. Foster, Thorsten Langner, Angus Malmgren, Adeline Harant, Cassandra Jensen, Diane G. O. Saunders, Joe Win, Hernán A. Burbano, Sophien Kamoun.

**Data curation:** Sergio M. Latorre, Vincent M. Were, Andrew J. Foster, Thorsten Langner, Daniel Croll, Joe Win.

**Formal analysis:** Sergio M. Latorre, Vincent M. Were, Andrew J. Foster, Thorsten Langner, Angus Malmgren, Adeline Harant, Sarai Reyes-Avila, Cassandra Jensen, Darren M. Soanes, Diane G. O. Saunders, Yukio Tosa, Daniel Croll, Kurt H. Lamour, Joe Win, Nicholas J. Talbot, Hernán A. Burbano, Sophien Kamoun.

**Funding acquisition:** Tofazzal Islam, Joe Win, Nicholas J. Talbot, Hernán A. Burbano, Sophien Kamoun.

**Investigation:** Sergio M. Latorre, Vincent M. Were, Andrew J. Foster, Thorsten Langner, Angus Malmgren, Adeline Harant, Cassandra Jensen, Lauren Ryder, Darren M. Soanes, Daniel Croll, Kurt H. Lamour.

**Methodology:** Sergio M. Latorre, Vincent M. Were, Andrew J. Foster, Thorsten Langner, Angus Malmgren, Adeline Harant, Cassandra Jensen.

**Project administration:** Joe Win, Nicholas J. Talbot, Hernán A. Burbano, Sophien Kamoun.

**Resources:** Sergio M. Latorre, Vincent M. Were, Andrew J. Foster, Thorsten Langner, Soichiro Asuke, Sarai Reyes-Avila, Dipali Rani Gupta, Weibin Ma, Nur Uddin Mahmud, Md. Shabab Mehebub, Rabson M. Mulenga, Abu Naim Md. Muzahid, Sanjoy Kumar Paul, S. M. Fajle Rabby, Abdullah Al Mahbub Rahat, Lauren Ryder, Ram-Krishna Shrestha, Suwilanji Sichilima, Pawan Kumar Singh, Alison R. Bentley, Yukio Tosa, Daniel Croll, Kurt H. Lamour, Tofazzal Islam, Batiseba Tembo, Joe Win.

**Software:** Sergio M. Latorre, Thorsten Langner, Ram-Krishna Shrestha.

**Supervision:** Joe Win, Nicholas J. Talbot, Hernán A. Burbano, Sophien Kamoun.

**Validation:** Sergio M. Latorre, Vincent M. Were, Andrew J. Foster, Thorsten Langner.

**Visualization:** Sergio M. Latorre, Vincent M. Were, Andrew J. Foster, Thorsten Langner.

**Writing – original draft:** Sergio M. Latorre, Vincent M. Were, Andrew J. Foster, Thorsten Langner, Joe Win, Nicholas J. Talbot, Hernán A. Burbano, Sophien Kamoun.

**Writing – review & editing:** Sergio M. Latorre, Vincent M. Were, Andrew J. Foster, Thorsten Langner, Joe Win, Nicholas J. Talbot, Hernán A. Burbano, Sophien Kamoun.

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
