## [Editor Report · Decision Letter 0]

3 Nov 2022

Dear Dr. Kamoun, 

Thank you for submitting your manuscript entitled "A pandemic clonal lineage of the wheat blast fungus" for consideration as a Discovery Report by PLOS Biology.

Your manuscript has now been evaluated by the PLOS Biology editorial staff, as well as by an academic editor with relevant expertise, and I am writing to let you know that we would like to send your submission out for external peer review.

Once your full submission is complete, your paper will undergo a series of checks in preparation for peer review. After your manuscript has passed the checks it will be sent out for review. To provide the metadata for your submission, please Login to Editorial Manager (https://www.editorialmanager.com/pbiology) within two working days, i.e. by Nov 05 2022 11:59PM.

Kind regards,

Paula

---

Senior Editor

PLOS Biology

---

## [Decision Letter · Decision Letter 1]

21 Dec 2022

Dear Sophien,

Thank you for your patience while your manuscript "A pandemic clonal lineage of the wheat blast fungus" went through peer-review at PLOS Biology. Your manuscript has now been evaluated by the PLOS Biology editors, an Academic Editor with relevant expertise, and by several independent reviewers. I'm handling this paper temporarily while my colleague Dr Paula Jauregui is out of the office.

In light of the reviews, which you will find at the end of this email, we are pleased to offer you the opportunity to address the comments from the reviewers in a revision that we anticipate should not take you very long. We will then assess your revised manuscript and your response to the reviewers' comments with our Academic Editor aiming to avoid further rounds of peer-review, although might need to consult with the reviewers, depending on the nature of the revisions.

IMPORTANT: Please address the following:

a) Please could you change the title to something a bit more declarative? We suggest something like: "Genomic surveillance identifies a pandemic clonal lineage of the wheat blast fungus"

b) Please attend to the requests from the reviewers.

c) Please ensure that you comply with our Data Policy requests; specifically, we need you to supply the numerical values underlying Figs 1ABC, 2ABC, 3C, 4A, S1, S2, S3, S4, S5, S6, S7AB, S8AB, S9, S10, S11, S12CD, S14 (some of these will be treefiles, I guess, rather than numbers), either as a supplementary data file or as a permanent DOI’d deposition. I note that some of these data may be in your GitHub deposition (https://github.com/Burbano-Lab/wheat-clonal-linage); if so, please clarify, and supply a DOI’d version (e.g. in Zenodo, Figshare, etc.)

d) Please cite the location of the data clearly in all relevant main and supplementary Figure legends, e.g. “The data underlying this Figure can be found in S1 Data” or “The data underlying this Figure can be found in https://doi.org/XXXX”

We expect to receive your revised manuscript within 6 weeks. Please email us (plosbiology@plos.org) if you have any questions or concerns, or would like to request an extension. 

**IMPORTANT - SUBMITTING YOUR REVISION**

*Resubmission Checklist*

*Published Peer Review*

*PLOS Data Policy*

*Blot and Gel Data Policy*

Sincerely,

Roli

Roland Roberts PhD

Senior Editor

PLOS Biology

rroberts@plos.org

on behalf of

Editor

PLOS Biology

REVIEWERS' COMMENTS:

Reviewer #1:

[identifies themself as Johanna Rhodes]

This paper addresses the threat of wheat blast disease, which threatens global food security, and includes work that I believe is incredibly novel on at the forefront of this field, by combining genomics and laboratory experiments. I only have a few comments that I believe, if addressed, would make this article even stronger.

For the introduction, could you expand on the sentence explaining the importance of the carrying the RWT4 gene, but not RWT3, and how this ties in with the PWT3/4 effectors?

Methods:

Some more clarity on the 84 diagnostic SNPs would be helpful; I looked at the referenced paper in the methods (article 7, "Emergence of wheat blast in Bangladesh was caused by a South American lineage of Magnaporthe oryzae"), yet I couldn't see how the 84 SNPs were derived. As such, a brief description describing how the genotypes were extracted, and what these diagnostic 84 SNPs are would be helpful, as at the moment I can't see how they are informative for discriminating between lineages and deriving ancestral origins. It might be that this has been done elsewhere, and hasn't been sufficiently referenced in this paper; but if not, benchmarking how these 84 SNPs are indeed informative in comparison to the whole genome, would aid this part of the paper considerably.

There is mention of missing sites being removed; including missing positions, but labelling them as such, can be informative. Missing positions could be missing due to sequencing/PCR error or removed doing the bioinformatics analysis; this does not mean the position is not there, just that you are not certain it's there to high confidence. Including missing positions, and setting them to missing would be much closer to the biology.

Finally, regarding the temporal analysis using BEAST: whilst HKY is less complex, this might not necessarily be the best fit for your data. A better approach would be to use bModelTest (under different combinations of demographic and clock scenarios) to assess which is the best model to use. The dates of emergence need confidence intervals (in the Results section), and the rates seem very small - how do these compare to other rates in fungi? I think increasing the MCMC chain length for convergence and altering the model/demographic would improve rates and dates.

All in all - this is a very good piece of research, and must represent a huge effort from all involved, and I'd be happy to see this published.

Reviewer #2:

[identifies themself as Ping Wang]

Compared to Magnaporthe oryzae pathotype Oryza which causes the devastating rice blast, the M. oryzae pathotype Triticum infects wheat, causing less known but economically important wheat blasts in geologically limited areas. However, due to world trade, the wheat blast has spread to South Asia and South Africa. Studies are urgently needed to monitor the transmission to avoid continued widespread impact on world food supplies. Here, Latorre and colleagues employed SNP analysis to examine the genotypes of various strains collected from the three continents to establish a single clonal linkage of the fungus. They also provided evidence demonstrating that the strains with the similar genetic backgrounds are controllable by host plants harboring the avirulent Rmg8 gene and by the fungicide Strobilurin. This is an excellent written study of significant importance. The conclusions are largely supported by experimental approaches and substantiated by statistical analysis. 

I have no additional specific comments other than suggesting that the excellent review article entitled "Intercontinental Jumps and Its Management Strategies" by P. K. Singh and colleagues be included in the References.

Reviewer #3:

Latorre et al. addresses a topic of broad interest and approaches the problem of introductions of potentially devastating crop pathogens from a number of important angles and represents a strong contribution to plant pathology. These angles include phylogenetic and population genetic analyses of a global collection of strains of the wheat blast fungus, genomic analyses of genes conferring virulence and antifungal drug resistance, tests of virulence and drug resistance, and mating ability. My only concern is about the claim of independence for the two introductions and my concern can be addressed by some straightforward additional analyses involving the current dataset and some additional discussion. I have put my comments below quotes from the manuscript, which are preceded by @. 

 @Wheat, the most important food crop,

Some more information is needed. When I Google crops, rice comes up as the number one food crop.

 @ following two independent introductions from South America

It is clear that there were two introductions, one to Bangladesh and one to Zambia, but it is not clear that they were independent. I see these possibilities. 1. One introduction from South America to Bangladesh and one from South America to Zambia. 2: One introduction from South America to Bangladesh and then an introduction from Bangladesh to Zambia. 3: One introduction from South America to Zambia and then an introduction from Zambia to Bangladesh. 4. The involvement of introductions to other, unsampled, regions of the globe. I do not think that you have enough data to settle on #1 and rule out the others. Number 4 is impossible to refute, but you should acknowledge it. Think about early publications on amphibian decline or Cryptococcus gatti that created scenarios to explain introductions that turned out to be wrong. You can determine if you have enough data to distinguish among scenarios numbers 1 - 3 by using trees constrained to reflect the scenarios and using likelihood ratio tests to see if your data are significantly more likely for one of them. My guess is that you won't be able to exclude both numbers 2 and 3. In which case, it would be best to consider the various scenarios and comment on how other data (historical records, etc.) support or refute each scenario. 

 @cause total crop failure (5)

Are there data on the effect of wheat blast on the production of wheat in Bangladesh or Zambia? Reference 5 is about the synchrony of crop loss and not about wheat blast losses. 

 @twin challenge of climate change and armed conflicts in major agricultural regions.

Is there a reference for climate change or armed conflict and wheat blast? Or, one about the two problems and agricultural production in general?

 @However, the genetic identity and origin of the causal agent of an African outbreak, first detected in Zambia in 2018, remains unknown (8).

The authors cite reference 11 later in their ms regarding recombined regions of the wheat blast genome and this bioRxiv article also addresses the origin of the Zambian isolates. If there were any way to publish both this ms and that of ref. 11 back-to-back, science would be better served. Note that Latorre et al. goes well beyond reference 11 in testing virulence and drug resistance.

 @Here, we show that the recent emergence of wheat blast in Asia and Africa was caused by a single clonal lineage of the wheat blast fungus closely related to South American isolates and that the outbreaks in Zambia and Bangladesh originated by independent introductions.

See above comment about the independence of the introductions. It is also not clear to me that the introductions need to be single. Couldn't several closely related but genetically different strains have been involved in the introductions?

 @The B71 lineage shows reduced genetic diversity in comparison with South American isolates although incipient sub-structuring can be noted between Zambian and Bangladeshi clusters (Fig. 2A, inset).

Figure 7a and 7b show that much of the genetic variation in the Zambian clade may be due to the genome rearrangements and the mini-chromosome. I wonder if the authors have analyzed the amount of genetic variation and the timing of introductions after removing the non-clonal variation? If so, did it alter the story about the timing of introductions? If not, it would be worth doing to see if it does alter the timing.

 @The B71 cluster is a clonal lineage.

This claim is solid.

 @The B71 clonal lineage has recently expanded with independent introductions in Zambia and Bangladesh.

See comments above.

 @These findings are consistent with the conclusions of a recent independent study (11).

The Liu et al. bioRxiv publication advances a good phylogenetic argument for two, independent introductions, because the Bangladesh isolates are on a branch with a Brazilian isolate at the base and the Zambian isolates are on a branch subtended by a Bolivian isolate. Again, likelihood ratio tests would show if there are enough data to support this scenario over the others.

 @we removed regions that disrupt the clonal pattern of inheritance (Fig. S6-S8)

More explanation is needed here. Having just told the reader that the spread is clonal, you need to let the reader know how there can be non-clonal elements in the genome. Figure S7 needs to be described in more detail. This figure also provides an argument in favor of independent introduction because the Zambian isolates and the Bolivian isolate seem to share a mini-chromosome that is absent in the Brazilian and Bangladeshian isolates. This point is also made in the Liu et al. bioRxiv article.

 @We scanned the available genomes and found that AVR-Rmg8 is conserved in all 36 isolates of the B71 clonal lineage even though the other 35 isolates of the Triticum lineage carry four diverse AVR-Rmg8 

 virulent alleles (eII, eII', eII'', eII''') that fully or partially evade immunity (Fig. 3A; Fig. S11) (16). B71 lineage isolates also lack the PWT4 effector, which is known to suppress AVR-Rmg8-elicited resistance 

 (17).

This section of the ms is solid.

 @These genome analyses predict that the B71 lineage isolates (AVR-Rmg8 positive, PWT4 negative) cannot infect wheat plants with the matching disease resistance gene Rmg8. To test this, we inoculated 14 

 B71 lineage isolates from Zambia and Bangladesh on wheat lines with and without the Rmg8 resistance gene (Fig. 3B; Fig. S12). Unlike a distinct South American isolate, none of these pandemic isolates could 

 infect Rmg8 wheat plants.

This section of the ms is solid.

 @Remarkably, all but one Brazilian isolate (12.1.181) of the 36 B71 lineage genomes carry the G1243C allele and are predicted to be strobilurin sensitive. We tested this by assaying B71 lineage isolates and 

 found that all tested 30 isolates are strobilurin sensitive (Fig. 4B-C; Fig. S13).

From the legend to Figure 4b-c, it seems that only the Zambian isolates were tested for antifungal resistance. If strain 12.1.181 has the genome of a strobilurin resistant strain, and it is basal to the Bangladesh clade, are all Bangladesh strains resistant? How did susceptible strains emerge from a population of resistant strains? This point raised the possibility of other populations of wheat blast in other global locations. Note in Figure 4b that there is some phylogenetic distance between the Bangladesh or Zambia populations and their closest South American relative.

All data appear to be available to the scientific community.

---

## [Editor Report · Decision Letter 2]

24 Feb 2023

Dear Dr Kamoun,

Thank you for the submission of your revised Discovery Report "Genomic surveillance uncovers a pandemic clonal lineage of the wheat blast fungus" for publication in PLOS Biology. On behalf of my colleagues and the Academic Editor, Joseph Heitman, I am pleased to say that we can in principle accept your manuscript for publication, provided you address any remaining formatting and reporting issues. These will be detailed in an email you should receive within 2-3 business days from our colleagues in the journal operations team; no action is required from you until then. Please note that we will not be able to formally accept your manuscript and schedule it for publication until you have completed any requested changes.

PRESS

Sincerely, 

Paula

---

Senior Editor

PLOS Biology
